Citation: *Molecular Systems Biology* 9:694
www.molecularsystemsbiology.com

# A competitive protein interaction network buffers Oct4-mediated differentiation to promote pluripotency in embryonic stem cells

Silvia Muñoz Descalzo[1,2,6], Pau Rué[3,4,6,7], Fernando Faunes[1,8], Penelope Hayward[1], Lars Martin Jakt[5,9], Tina Balayo[1], Jordi Garcia-Ojalvo[3,4] and Alfonso Martinez Arias[1,*]

[1] Department of Genetics, University of Cambridge, Cambridge, UK, [2] Biology and Biochemistry Department, University of Bath, Bath, UK, [3] Department of Experimental and Health Sciences, Universitat Pompeu Fabra, Barcelona Biomedical Research Park, Barcelona, Spain, [4] Departament de Física i Enginyeria Nuclear, Universitat Politècnica de Catalunya, Terrassa, Spain and [5] Stem Cell Biology Group, Riken Center for Developmental Biology, Kobe, Japan
[6]These authors contributed equally to this work
[7]Present address: Department of Genetics, University of Cambridge, Cambridge, UK
[8]Present address: Facultad de Ciencias Biologicas, Pontificia Universidad Católica de Chile, Avda. Libertador Bernardo OHiggins 340, Santiago, Chile
[9]Present address: Department of Systems Medicine, Mitsunada Sakaguchi Laboratory, Keio University School of Medicine, Tokyo, Japan.
* Corresponding author. Department of Genetics, University of Cambridge, Downing Street, Cambridge CB2 3EH, UK. Tel.: + 44 (0)1223 766742; Fax: + 44 (0)1223 333992; E-mail: ama11@hermes.cam.ac.uk

**Pluripotency in embryonic stem cells is maintained through the activity of a small set of transcription factors centred around Oct4 and Nanog, which control the expression of 'self-renewal' and 'differentiation' genes. Here, we combine single-cell quantitative immunofluorescence microscopy and gene expression analysis, together with theoretical modelling, to investigate how the activity of those factors is regulated. We uncover a key role for post-translational regulation in the maintenance of pluripotency, which complements the well-established transcriptional regulatory layer. Specifically, we find that the activity of a network of protein complexes involving Nanog, Oct4, Tcf3, and β-catenin suffices to account for the behavior of ES cells under different conditions. Our results suggest that the function of the network is to buffer the transcriptional activity of Oct4, which appears to be the main determinant to exit pluripotency. The protein network explains the mechanisms underlying the gain and loss of function in different mutants, and brings us closer to a full understanding of the molecular basis of pluripotency.**
*Molecular Systems Biology* **9**: 694; published online 8 October 2013; doi:10.1038/msb.2013.49
*Subject Categories:* simulation and data analysis; differentiation & death
*Keywords:* β-catenin; mathematical modelling; Oct4; pluripotency; protein network

## Introduction

Embryonic stem (ES) cells are clonal cell populations derived from mouse pre-implantation embryos that, upon differentiation, give rise to derivatives of all the lineages of an organism and, under certain culture conditions, propagate this capacity, that is, they are pluripotent (Bradley *et al*, 1984; Smith, 2001). Genetic and molecular analyses have revealed a small network of transcription factors centred around the activity of Pou5f1/ Oct4 (Oct4) and Nanog that is necessary and sufficient to establish and maintain pluripotency in ES cells, as well as to drive the reprogramming of differentiated cells into this state (Silva and Smith, 2008; Young, 2011).

Analysis of the expression of Oct4 and Nanog in standard self-renewal conditions, for example, Serum and LIF (S + L), reveals a high degree of heterogeneity in gene expression and protein levels (reviewed in Martinez Arias and Brickman, 2011). This is particularly obvious for Nanog, which exhibits a wide range of expression levels that are functionally significant: populations of

cells with high levels of Nanog have a higher probability to self-renew while with low levels are prone to differentiate (Chambers *et al*, 2007; Kalmar *et al*, 2009; Miyanari and Torres-Padilla, 2012; Navarro *et al*, 2012). In contrast, the expression of Oct4 is tightly regulated, with deviations of Oct4 levels from a set range of values resulting in differentiation (Niwa *et al*, 2000, 2002). When ES cells are grown in the presence of PD03, an inhibitor of the ERK kinase MEK, and Chiron, an inhibitor of GSK3 (2i conditions), they exhibit high levels of Nanog and Oct4 and do not differentiate; this state has been termed as the 'ground state' of pluripotency (Ying *et al*, 2008; Wray *et al*, 2010). The effects of 2i are thought to be mediated by the convergent activity of the inhibitors: PD03 increases the frequency of Nanog expression (Lanner *et al*, 2010; Miyanari and Torres-Padilla, 2012) while Chiron acts on β-catenin to neutralize the repressing activity of Tcf3 on the pluripotency network (Wray *et al*, 2011; Martello *et al*, 2012; Shy *et al*, 2013). Consistent with this interpretation, Fgf-driven ERK signalling, loss of β-catenin, or increases in Tcf3, result in loss of pluripotency and differentiation (Pereira *et al*,

2006; Kunath *et al*, 2007; Lyashenko *et al*, 2011; Wray *et al*, 2011; Yi *et al,* 2011). Despite these observations, the molecular mechanisms that establish and maintain pluripotency remain open to discussion.

Recent results have revealed the existence of protein complexes involving β-catenin, Nanog, Oct4, and Tcf3, which are associated with pluripotency in ES cells (Wang *et al*, 2006; Takao *et al*, 2007; Abu-Remaileh *et al*, 2010; Kelly *et al*, 2011; Ding *et al*, 2012; Costa *et al*, 2013; Faunes *et al*, 2013; Zhang *et al*, 2013). Here, we use a combination of experiments and modelling to show that the relative amounts of these complexes are established by a dynamic competition between their individual elements, and suggest that their function is to buffer the differentiation-promoting activity of Oct4. Thus, the stability of the pluripotent state of ES cells is likely to be determined by the dynamic balance of the protein complexes, and can be expected to be associated with the ratios between their components.

# Results

## Oct4/Nanog complex formation as the molecular basis of pluripotency: the NOC model

Quantitative immunofluorescence (QIF) of single mES cells shows that pluripotency is associated with specific ratios, rather than absolute levels, of Oct4 and Nanog protein expression (Muñoz Descalzo *et al*, 2012). These ratios manifest themselves as correlations between the levels of Oct4 and Nanog that increase when ES cells are cultured in 2i conditions (Figure 1A and B). A salient feature of the joint distribution of Oct4 and Nanog levels in single cells is a sharp boundary below which no cells can be found in self-renewing conditions (Muñoz Descalzo *et al*, 2012, see also Figure 1A). This boundary represents a lower limit to the levels of Oct4 that a cell can have for a given amount of Nanog, and its tilted layout indicates that this limit varies with the latter. Current mathematical models of pluripotency focus on transcriptional control and cannot reproduce such a constraint. For example, an excitable transcriptional system driven by fluctuations in Nanog expression (Kalmar *et al*, 2009; see also Supplementary Figure S1) captures the dynamical features of Nanog gene expression in a pluripotent cell population, but is unable to reproduce the observed correlations of protein expression (see Supplementary Figure S1C and Supplementary information for additional considerations). This raises the possibility that post-transcriptional and post-translational mechanisms, which have been shown to be associated with ES cells (Sampath *et al*, 2008; Buckley *et al*, 2012; Faunes *et al*, 2013; Sanchez-Ripoll *et al*, 2013), contribute to the regulation of pluripotency. In support of this, treatment of mES cells with a proteasome inhibitor for 2 h increases the levels of Oct4 and Nanog, reducing the heterogeneity in Nanog expression (Figure 1C; Supplementary Figure S2). These observations have led us to consider the role that post-translational regulatory mechanisms have in the regulation of the levels of Nanog and Oct4 in ES cells.

To test the relevance of post-translational regulation in pluripotency, we first consider a minimal network involving Oct4 and Nanog (Figure 1F), on the assumption that the correlations between these two proteins result from the formation of a complex (O:N) that has been previously described experimentally (Wang *et al*, 2006; Zhang *et al*, 2007; van den Berg *et al*, 2010; Ding *et al*, 2012; Fidalgo *et al*, 2012). In this minimal model (NOC model, for **N**anog-**O**ct4-**C**omplex), we assume that Oct4 and Nanog exist in one of the two forms: either free or bound together in a complex. We do not exclude the possibility that the free forms of Nanog and Oct4 interact with other proteins to exert additional functions (see below). This model aims to describe the stochastic dynamics of Oct4 and Nanog expression and translation without relying on any specific transcriptional regulation (see Supplementary information). The model surmises that in mES cells Nanog is transcribed in infrequent bursts, as observed experimentally (Figure 1H and I; Supplementary Figure S4A and C; Miyanari and Torres-Padilla, 2012; Navarro *et al*, 2012). The parameters associated with the transcriptional interactions are chosen to fit these expression data. The model also assumes that free Nanog is degraded at a rate faster than free Oct4, but that it is stabilized by forming a complex with Oct4. Stochastic simulations of this system for basal parameters representing S + L conditions reproduce, to a good approximation, the experimentally observed protein distributions and correlations for Nanog and Oct4 (Figure 1D and G).

We next ask whether the model is able to reproduce the correlations between Nanog and Oct4 observed when cells are cultured in 2i. We implement this using the observation that in this condition there is an increase in the number of cells with higher Nanog mRNA levels (Figure 1H and I) and represent this by continuously feeding the system with newly synthesized Nanog molecules (Supplementary Figure S4B and D). Assuming a sufficiently high affinity of Oct4 for Nanog, the high levels of Nanog in 2i (Supplementary Figure S4E) will drive most of the available Oct4 into the complex and ensure that only levels close to a given ratio (corresponding to the lower boundary of the scatter plot in S + L) are occupied (Figure 1E and G). Simulations of the model reproduce the observation that the ground state is not associated with a particular level of Oct4 and Nanog, but rather with a continuum of levels that lie along a straight line in the Nanog-Oct4 plane (Muñoz Descalzo *et al*, 2012; Supplementary Figure S4G and H).

While this minimalistic model can account for the correlations between Oct4 and Nanog in the ground state, it cannot explain some important observations. In particular, it does not include β-catenin, whose levels have been shown to have a significant role in the regulation of the pluripotency network (Lyashenko *et al*, 2011; Wray *et al*, 2011; Faunes *et al*, 2013). Most significantly, according to the model, the absence of Nanog should lead to elevated levels of free Oct4, which experimentally has been shown to promote differentiation, and yet Nanog mutant cells remain pluripotent (Chambers *et al*, 2007). This suggests that additional elements and interactions need to be incorporated into the model.

## A protein interaction network involving Oct4, Nanog, and β-catenin underlies naïve pluripotency: the TBON model

Molecular analyses have revealed a dual role for β-catenin in the maintenance of pluripotency: alleviating the repressive

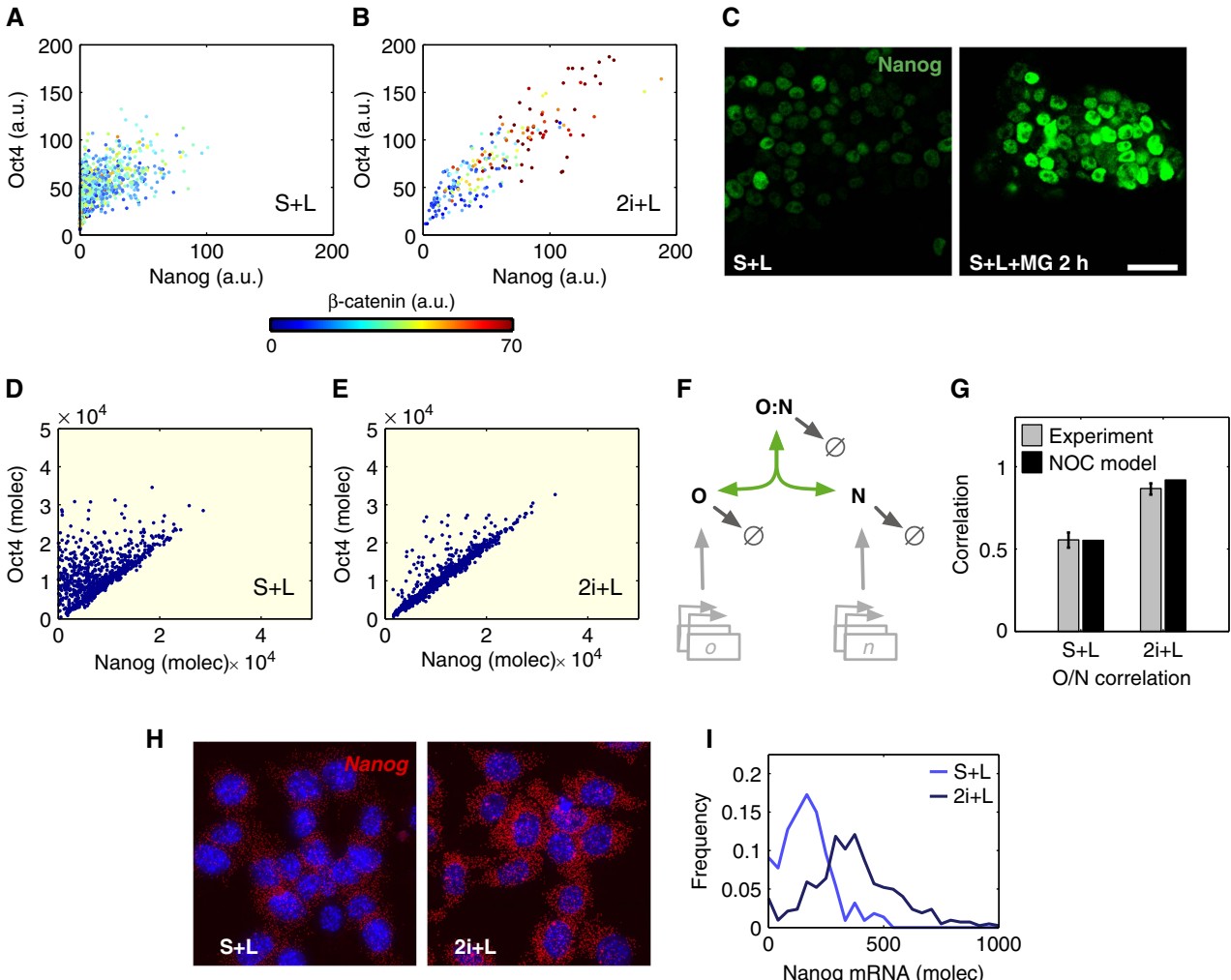

**Figure 1** The NOC model. (**A**, **B**) Scatter plots showing Nanog (*x* axis), Oct4 (*y* axis), and β-catenin (heat map, bottom bar) levels, in fluorescence arbitrary units (a.u.) here and in subsequent similar graphs, in single E14Tg2A cells under standard Serum + LIF (S + L) conditions (**A**) and 2i + LIF (2i + L) conditions (**B**). Each dot represents the levels in a single cell. (**C**) Representative confocal images of Nanog protein expression in E14Tg2A cells grown in S + L (left panel) and treated with the proteasome inhibitor MG132 for 2 h (S + L + MG, right panel) before fixation. Scale bar: 50 μm. (**D**, **E**) Numerical simulation results in standard S + L (**D**) and 2i + L (**E**) conditions. Blue dots represent protein levels of cells sampled from the model simulations. For simplicity, here and in subsequent figures, we include a light vanilla background when showing modelling results. (**F**) Scheme of the core protein NOC network, including Nanog (N), Oct4 (O), and the Oct4:Nanog (O:N) complex. (**G**) Pearson's correlation coefficient between Oct4 and Nanog from the experimental data (gray bars), and the simulations (black bars), in cells cultured in S + L and 2i + L. The error bars in the experimental data indicate 95% CI of bootstrapping distribution, here and in similar subsequent graphs. (**H**) Representative RNA-FISH images of EB5 cells grown in S + L (left) or 2i + L (right) and hybridized against Nanog mRNA using a Cy5-labeled probe (red channel), nuclei are shown in blue. (**I**) Distributions of Nanog mRNA molecules number in cells grown in S + L (light-blue line) and 2i + L (dark-blue line) obtained from the hybridization experiment shown in (**H**).

activity of Tcf3 on Nanog (Wray *et al*, 2011; Martello *et al*, 2012; Zhang *et al*, 2013) and through a complex with Oct4 (Takao *et al*, 2007; Abu-Remaileh *et al*, 2010; Kelly *et al*, 2011; Ding *et al*, 2012; Faunes *et al*, 2013). To add these interactions to the NOC model, we first analyzed the experimental distributions of β-catenin in relation to those of Oct4 and Nanog at the level of single cells (Figures 1A, B, and 2A).

Under standard growth conditions, there is no clear correlation between the levels of β-catenin and Nanog (β/N), and the correlation between β-catenin and Oct4 (β/O) is weak (Figures 1A and 2E, left gray bars). In contrast, when cells are cultured in 2i, the β/N and β/O correlations increase significantly, indicating that they represent features of ground-state pluripotency (Muñoz Descalzo *et al*, 2012)

(Figures 1B and 2E, right gray bars). As in the NOC model, we consider that these correlations might be caused in part by the existence of constrained interactions of β-catenin with Oct4 and Nanog, in the form of protein complexes that have been detected (Wang *et al*, 2006; Takao *et al*, 2007; Zhang *et al*, 2007; Abu-Remaileh *et al*, 2010; van den Berg *et al*, 2010; Kelly *et al*, 2011; Ding *et al*, 2012; Faunes *et al*, 2013; Zhang *et al*, 2013), and use this to expand the NOC model. The resulting model, TBON (for Tcf3, β-catenin, Oct4 and Nanog) (Figure 2B), considers the existence of four protein complexes, O:N, β:O, β:T, and β:O:N, which compete for their individual components, and includes assumptions for which there is experimental evidence. These assumptions can be summarized as follows:

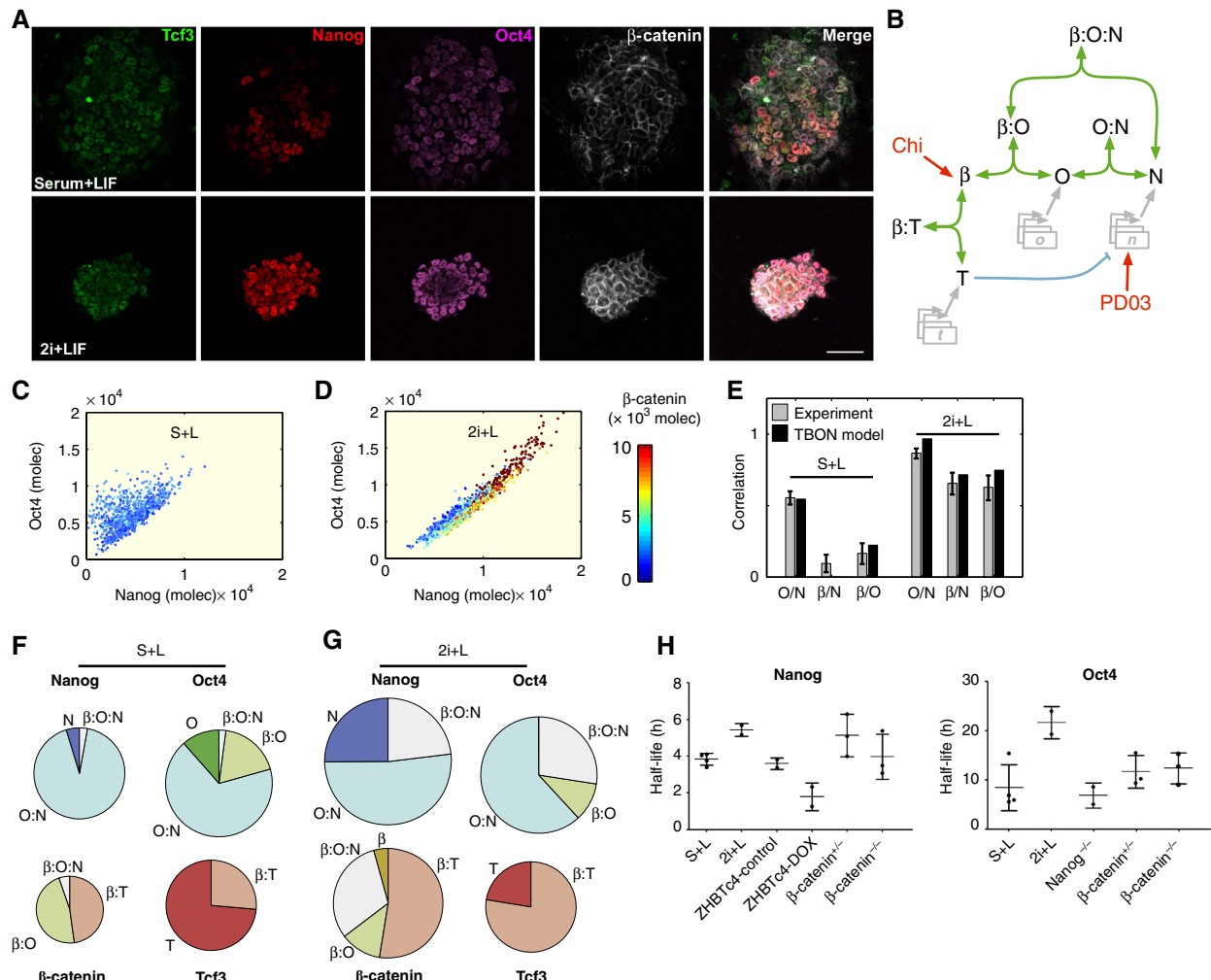

**Figure 2** The TBON model. (**A**) Representative confocal images of E14Tg2A cells stained for Tcf3 (green), Nanog (red), Oct4 (magenta), and total β-catenin (white) grown in S + L (upper panels) and 2i + L (lower panels). Scale bar: 50 μm. (**B**) Scheme of the core protein TBON network. See text for details. (**C**, **D**) Numerical simulation of the model for Nanog (*x* axis), Oct4 (*y* axis), and β-catenin (heat map) levels in standard S + L (**C**) and 2i + L (**D**) conditions; compare with the experimental data shown in Figure 1A and B. Each dot is a cell sampled from the model simulations. (**E**) Pearson's correlation coefficient between Oct4 and Nanog (O/N), β-catenin and Nanog (β/N), and β-catenin and Oct4 (β/O) from the experimental data (gray bars) and from the TBON simulations (black bars), in cells cultured in S + L and 2i + L. (**F**, **G**) Pie charts showing the relative pools of Nanog, Oct4, β-catenin, and Tcf3 found as free molecules (dark colors) or as part of a complex (pale colors), as determined by the TBON model in cells grown in S + L (**F**) or 2i + L (**G**). The area of each circle is proportional to the total amount of each respective molecule and so are the circle sectors to the corresponding free proteins and complexes. (**H**) The plot shows the mean and standard deviation of 2–4 independent experiments of Nanog and Oct4 half-lives in the indicated genotypes and culture conditions.

(i) Tcf3 inhibits Nanog expression via a direct transcriptional interaction (Pereira *et al*, 2006; Yi *et al*, 2008; Martello *et al*, 2012; Shy *et al*, 2013; Zhang *et al*, 2013).

(ii) The impact of 2i on the network has two separate effects: (a) an increase in the levels of β-catenin as a consequence of Chiron and (b) an increase in the levels of Nanog expression as a consequence of PD03 (Lanner *et al*, 2010; Wray *et al*, 2010; Wray *et al*, 2011; Luo *et al*, 2012; Miyanari and Torres-Padilla, 2012).

(iii) The model considers the existence of three binary complexes: Oct4 with Nanog (O:N, Wang *et al*, 2006; Zhang *et al*, 2007; van den Berg *et al*, 2010; Fidalgo *et al*, 2012); β-catenin with Oct4 (β:O, Takao *et al*, 2007; Abu-Remaileh *et al*, 2010; Kelly *et al*, 2011; Ding *et al*, 2012; Faunes *et al*, 2013); and β-catenin and Tcf3 (β:T, Yi *et al*,

2011; Shy *et al*, 2013). Experimentally, we could detect a small amount of Nanog in the β:O complex which prompted us to include a β-catenin:Oct4:Nanog (β:O:N) ternary complex in the model.

(iv) There is a hierarchy in the half-lives of the elements of the network, and also in the stabilities of the complexes, that can be arranged from less to more stable as: Nanog < Oct4 < O:N < T, β:T < β:O:N, β:O, and β. This hierarchy of degradation rates is compatible with our experimental observations of the stability of the different protein species of the network (Figure 2H; Supplementary Figure S3).

(v) The total amount of β-catenin in the system is unlimited, but only a fraction of it is available to the pluripotent protein network. The available amount of β-catenin

varies from cell to cell, as we see experimentally (Figure 1A; Supplementary Figure S6C) and allows us to introduce a dynamic intercellular heterogeneity in β-catenin levels.

In the model, the activity of the individual factors is altered when they become part of a complex; for instance, Tcf3 is unable to repress Nanog expression when in complex with β-catenin (β:T) (Shy *et al*, 2013; Zhang *et al*, 2013). Stochastic simulations of the TBON model (see Supplementary information for details) reproduce the experimental protein distributions and correlations among all the elements of the extended network in different culture conditions (Figure 2C–E and compare Supplementary Figure S6A–D with E–H). Most significantly, this model, and its underlying set of interactions, provides a framework to understand the dynamics of the key proteins required for the maintenance of pluripotency, which ultimately relies on the relative levels of Tcf3, β-catenin, Oct4, and Nanog, found as either part of a complex or as free molecules in individual cells. The model shows that competition between the complexes for their common components leads to a balanced equilibrium that is dependent on the culture conditions (Figure 2F and G; Supplementary Figure S6L and M). The key element of the competition is the association and dissociation dynamics of the different complexes, together with the differential degradation kinetics of the different species. A comparison between Figure 2F and G shows that the β:O, β:T, and β:O:N complexes should be found more readily in 2i conditions than in S + L, whereas more 'free' Oct4 and Tcf3 will exist in S + L than in 2i, as it is found experimentally (Faunes *et al*, 2013). As a result, Nanog and Oct4 should be more stable in 2i, where their expression correlates, which is what we find experimentally (Figure 2H). The model allows a detailed dissection of the effects of each of the inhibitors of 2i, thus helping us to understand how the effects of Chiron and PD03 combine to maintain pluripotency (Supplementary Figure S8).

We surmise that the main function of the complexes is to stabilize ground-state pluripotency through the attenuation of the activity of Oct4, represented by free Oct4, which has been shown to lead to differentiation when in excess (Niwa *et al*,

2000; Nishiyama *et al*, 2009, 2013; Karwacki-Neisius *et al*, 2013; Radzisheuskaya *et al*, 2013). A bioinformatic analysis of the targets of Oct4 and Nanog supports this contention, and shows that although Oct4 and Nanog bind to many common sites in the genome, their effects on gene regulation are not the same (Figure 3; Supplementary Figure S9). Jointly regulated genes, which might be associated with high and correlated (balanced) levels of Nanog and Oct4, are mostly associated with pluripotency (Figure 3A). On the other hand, the unique targets of Nanog, and most clearly Oct4, include a number of transcription factors and signalling molecules associated with the emergence of germ layers and body patterning (Hox, Sox, Smad, Fgf, Wnt, Nodal signalling; see Figure 3B and Supplementary Figure S9, and also Niwa *et al*, 2005; Nishiyama *et al*, 2009, 2013; Loh and Lim, 2011; Iwafuchi-Doi *et al*, 2012). This analysis supports the notion that a main function of the protein complexes is to buffer the amount of Oct4 available to trigger differentiation, thus maintaining the activity of the pluripotency network.

## State of pluripotency in the absence of the protein network components

The results and model presented here create a framework to analyze mutants, gain some insights into the molecular basis for their phenotypes, and thereby on the mechanisms underlying the maintenance of pluripotency in wild-type conditions. For example, Nanog is a key element of the ES cell regulatory network but, surprisingly, while it is absolutely required for the establishment of pluripotency (Silva *et al*, 2008, 2009; Theunissen *et al*, 2011), its removal in an established ES cell population only affects the frequency of differentiation of the mutant cells but does not abolish pluripotency (Figure 4A; Supplementary Figure S10; Chambers *et al*, 2007). The model predicts that in the absence of Nanog there will be an increase in the levels of β-catenin that will bind Oct4 and that the increase in the β:O complex is what maintains, albeit unstably, pluripotency in these cells (compare Figure 2F and 4B, and see also Supplementary Figure S11). Nanog mutant cells, as predicted by the model (Supplementary Figure S11), exhibit

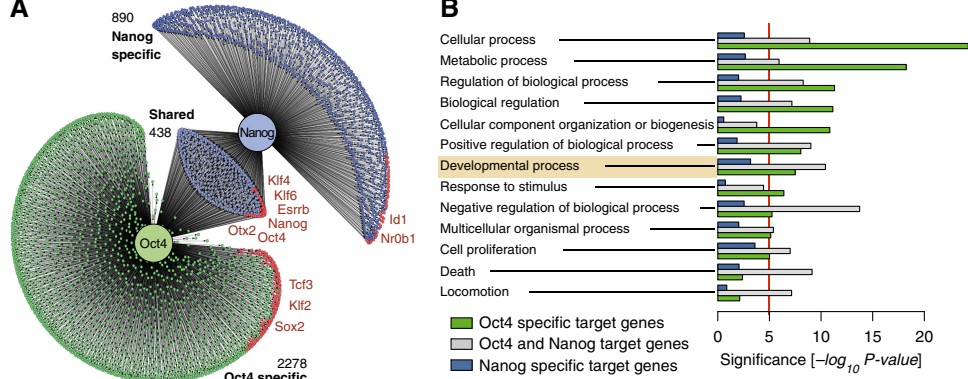

**Figure 3** Biological function of Oct4 and Nanog transcriptional targets. (**A**) Schematic network showing the common and specific target genes of Nanog and Oct4. Red nodes indicate target genes with known transcription factor activity. Some factors of special interest are highlighted in red and annotated in Supplementary Figure S9. See Supplementary information for details. (**B**) Significance, measured as the exponent of the *P*-value, of the Gene Ontology Biological Process terms for common and specific target genes of Nanog and Oct4. For details, see Supplementary information.

an increase in the levels of total Oct4 and β-catenin (Figure 4C) as well as in their correlation (Figure 4D), which indicates that the frequency of differentiation is very sensitive to the levels of β-catenin. Consistent with this prediction, we observe that the pluripotency of the Nanog mutant cells is completely dependent on Chiron (Figure 4A; Supplementary Figure S10). As Nanog is the least stable component of the network, our framework suggests that in its absence, the stability of the other components should not be significantly altered. This is precisely what we observe: Oct4 half-life does not change in the absence of Nanog (Figure 2H).

Our model is also able to reproduce and provide a molecular understanding of the phenotype of β-catenin mutant ES cells, which can be unstably maintained in 2i + LIF (Wray *et al*, 2011; Faunes *et al*, 2013). Although the levels of total Oct4 and Nanog are low in the absence of β-catenin (Supplementary Figure S12A and B; Faunes *et al*, 2013), the O/N correlation in these cells is not very different from that of the parental cell line (Supplementary Figure S12K) (Muñoz Descalzo *et al*, 2012), which is in agreement with the notion that a high O/N correlation is a central feature of pluripotency. The model reproduces this situation (Figure 5A; Supplementary Figure S12E–H and K) and suggests an explanation for it: the absence of β-catenin will increase the levels of Tcf3 (Figure 5B) that, in turn, will inhibit Nanog expression, lowering the amount of the O:N complex, thus contributing to the instability of the mutant cells. Our framework predicts that if we decrease the

levels of β-catenin, the half-life of Oct4 decreases by 40% which is compatible with experimental observations. On the other hand, the half life of Nanog is predicted to decrease by 16%, which might not be experimentally detectable (β-catenin[+/−] cells; Figure 2H). Interestingly, we do not observe a further decrease in the complete absence of β-catenin, suggesting that the stability of only a fraction of Oct4 is regulated by β-catenin. All together, these results provide an explanation for why β-catenin, although not absolutely necessary for self-renewal (Lyashenko *et al*, 2011; Wray *et al*, 2011; Rudloff and Kemler, 2012), is essential for the robustness of pluripotency. Consequently, β-catenin mutant cells exhibit a higher rate of differentiation, as has been observed (Wray *et al*, 2011; Faunes *et al*, 2013).

The ability of the model to reproduce and explain mutants with weakened pluripotency is important, but there are also mutants, in particular Tcf3 mutants, that have been suggested to exhibit increased pluripotency (Pereira *et al*, 2006; Yi *et al*, 2008; Guo *et al*, 2011). It is thought that the pluripotency of Tcf3 mutants is due to increased levels of Nanog, but we also observe a decrease in the levels of Oct4 and β-catenin, and a collapse of the O/N correlation, as cells with low O/N ratio cross the lower boundary (Figure 5C; Supplementary Figure S14A–F; Muñoz Descalzo *et al*, 2012). Introducing these experimental observations in the TBON model (see Supplementary information), we can reproduce the measured decrease in O/N correlation, suggesting that the excess Nanog

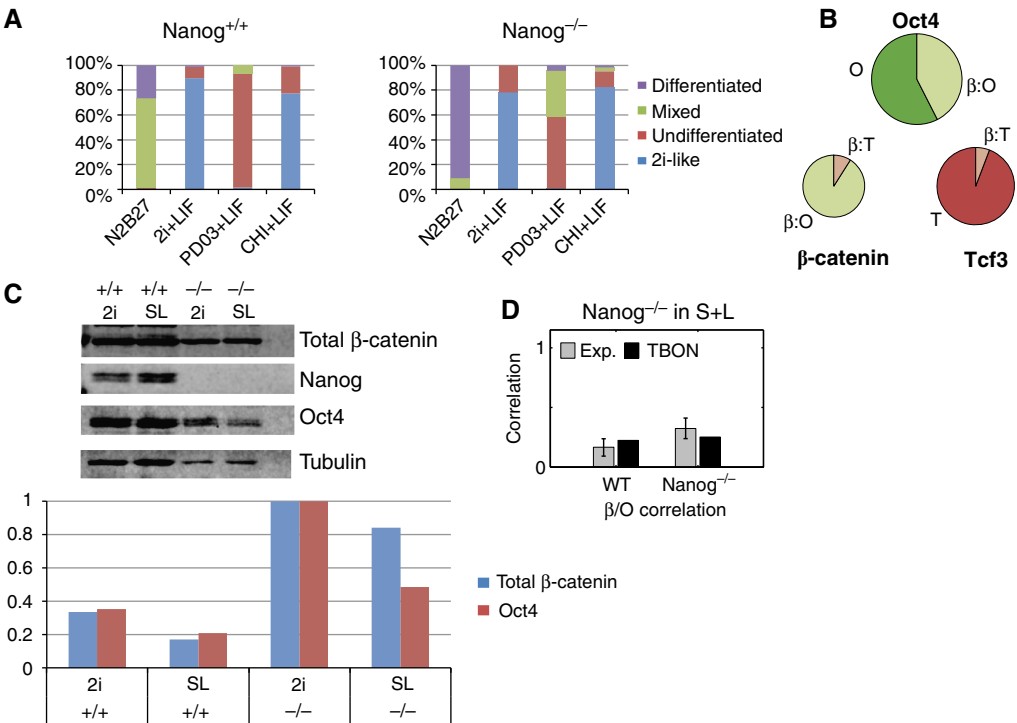

**Figure 4** Analysis of Nanog mutant cells. (**A**) Distribution of colony forming assays for Nanog[+/+] and Nanog[−/−] cells under the indicated conditions. Colonies were assayed for alkaline phosphatase (a marker for pluripotency). The relative percentages of the different colony types are shown. For a description of the colony types, see Supplementary Figure S10. (**B**) Pie charts showing the relative pools of Oct4, β-catenin, and Tcf3 found as free molecules (dark colors) or as part of a complex (pale colors) as determined by the TBON model in the absence of Nanog. (**C**) Western blot analysis of Nanog[+/+] and Nanog[−/−] cells grown in either S + L or 2i + L for at least two passages before lysis. The panel below shows the quantifications of western blots. Western blot membranes were scanned in the Oddysey System. The intensity of each band was normalized to tubulin and to the levels of β-catenin or Oct4 in Nanog mutant cells grown in 2i + L. (**D**) Pearson's correlation coefficient between Oct4 and Nanog from the experimental data (gray bars) and from the TBON simulations (black bars), in wild-type (WT) and Nanog mutant (Nanog[−/−]) cells cultured in S + L.

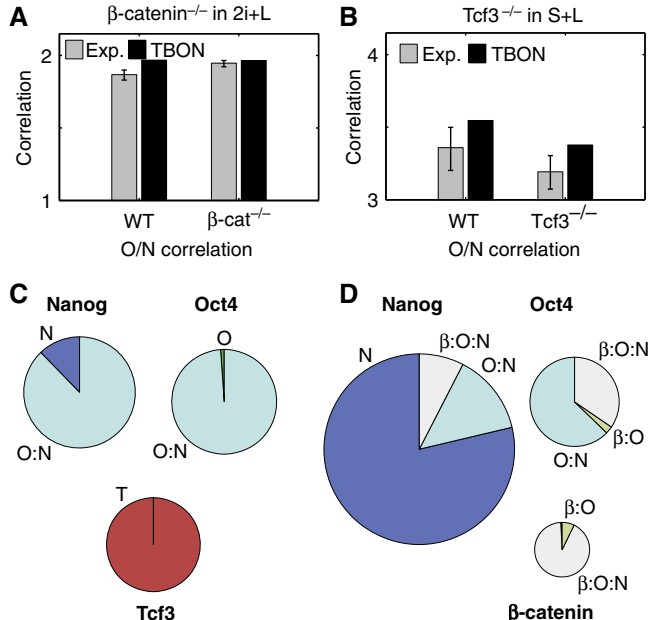

**Figure 5** Analysis of β-catenin and Tcf3 mutant cells. (**A**) Pearson's correlation coefficient between Oct4 and Nanog from the experimental data (gray bars) and from the TBON simulations (black bars) in wild-type (WT) and β-catenin mutant (β-catenin$^{-/-}$) cells cultured in 2i + L. (**B**) Pie charts showing the relative pools of Nanog, Oct4, and Tcf3 found as free molecules (dark colors) or as part of a complex (pale colors) as determined by the TBON model in the absence of β-catenin and in 2i + L conditions. (**C**) Pearson's correlation coefficient between Oct4 and Nanog from the experimental data (gray bars) and from the TBON simulations (black bars), in wild-type (WT) and Tcf3 mutant (Tcf3$^{-/-}$) cells cultured in S + L. (**D**) Pie charts showing the relative pools of Nanog, Oct4, and β-catenin found as free molecules (dark colors) or as part of a complex (pale colors) as determined by the TBON model in the absence of Tcf3.

quenches any excess Oct4 (Figure 5D; Supplementary Figure S14G–M). The absence of free Oct4 molecules in the Tcf3 mutant cells would explain the delayed differentiation of Tcf3 mutant cells (Pereira *et al*, 2006; Yi *et al*, 2008).

Finally, we examined the response of the TBON circuit to perturbations in Oct4. Loss of Oct4 results in the loss of pluripotency (Niwa *et al*, 2000, 2002) and our model suggests that this is due to a breakdown in the balance of protein complexes. The result of the new steady state is a larger pool of free Nanog, which is subject to fast degradation (Figure 6A; Figure 2H; see also Supplementary Figure S15). This can also be seen in Figure 6B and C, which compares the experimental and numerical observations. This decrease in the levels of Nanog emulates the situation of the Nanog null cells, which are more prone to differentiation (Figure 4A and Supplementary Figure S10, see also Chambers *et al*, 2007). Conversely, overexpression of Oct4 will alter the O/N ratio, freeing up Oct4 (Figure 6D and F; Supplementary Figure S16F) and leading to the expression of its unique targets and therefore to differentiation, as observed experimentally (Niwa *et al*, 2000; Nishiyama *et al*, 2009). The excess of Oct4 can be buffered by either excess Nanog (Theunissen *et al*, 2011) or β-catenin (Faunes *et al*, 2013; Supplementary Figure S16F), leading to O:N, β:O, and β:O:N complexes (Figure 6D) that will suppress Oct4-driven differentiation.

## Discussion

We have shown that a competitive network of protein complexes involving Oct4, Nanog, Tcf3, and β-catenin can

account for the major characteristics of pluripotency in mouse ES cells. At the centre of this network is the transcriptional activity of Oct4, whose levels are crucial for the maintenance of pluripotency and differentiation (Niwa *et al*, 2000, 2002; Nishiyama *et al*, 2009). We suggest that the Oct4 levels are dynamically buffered through the interactions of three protein complexes (O:N, β:O, and β:O:N) that compete for their common elements. The result of this competition is that the levels of Oct4 are restricted to a range that promotes pluripotency though fluctuations in those levels lead to cells with higher differentiation-promoting potential. The variety of experimental conditions and cell lines analyzed here, together with the expression profiles and protein stability data obtained, provide us with strong constraints for establishing the region of parameter space in which our model operates. Yet, a systematic sensitivity analysis (Supplementary Figure S7) shows that the model is robust to the parameters chosen. A prediction of our model that the lower the levels of Oct4, the higher the degree of pluripotency, finds support on recent experiments that show that cells with low stable levels of Oct4 are unable to exit pluripotency (Karwacki-Neisius *et al*, 2013; Radzisheuskaya *et al*, 2013).

Genetic studies have revealed other proteins involved in the regulation of pluripotency, such as Sox2, Tbx3, Klf4, and Essrb (Niwa *et al*, 2009; Festuccia *et al*, 2012; Martello *et al*, 2012). Within our model, one can consider these factors as contributors to the stability and dynamics of the central network that we have defined here. In particular, recent work has established a critical role of Essrb, which is repressed by Tcf3 and has the ability to substitute for Nanog function (Festuccia *et al*, 2012; Martello *et al*, 2012). Esrrb

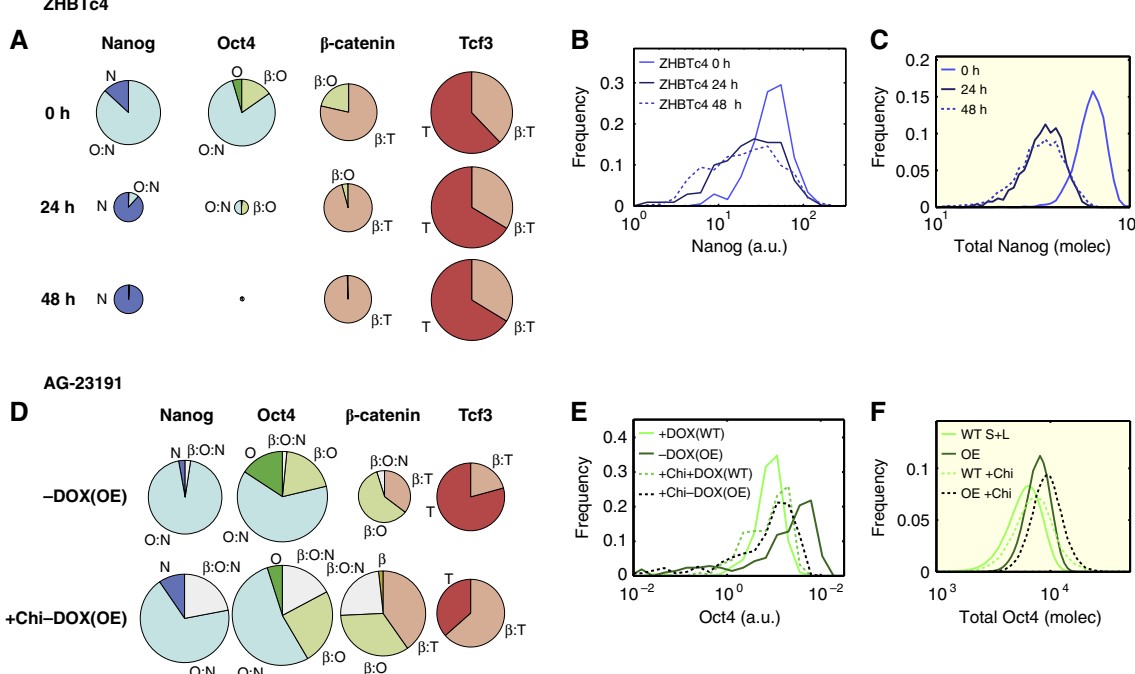

**Figure 6** Analysis of the pluripotent state upon perturbation of Oct4 levels. (**A**) Pie charts showing how the relative pools of Nanog, Oct4, β-catenin, and Tcf3 evolve over time upon removing Oct4 from the system for 0, 24, or 48 h as predicted by the model. (**B**) Distributions of Nanog in ZHBTc cells, untreated (light blue line, 0 h) and treated with Doxycycline, which represses Oct4 expression, for 24 h (dark-blue line) and 48 h (dashed line). Fluorescence levels (grayscale) were quantified for each individual cell, binned in logarithmically spaced classes (x axis); the frequency of each bin is shown on the y axis here and in similar graphs. (**C**) Nanog distributions exhibited by the TBON model, considering a wild-type situation in S + L (light-blue line, 0 h) and after stopping Oct4 production for 24 h (dark-blue line) and 48 h (dashed line). (**D**) Pie charts showing how the relative pools of Nanog, Oct4, β-catenin, and Tcf3 change upon Oct4 overexpression and β-catenin stabilization. (**E**) Oct4 distributions in AG-23191 cells grown in S + L + Doxycycline (light-green line, + DOX, WT) or removing Doxycycline for 24 h (dark-green line, − DOX, OE) which induces the overexpression of Oct4. The effect of β-catenin stabilization using Chiron was also analysed (dashed lines: light green for the wild-type, and dark green for the overexpressing conditions). The quantifications procedure was described in (**B**). (**F**) Oct4 distributions exhibited by the TBON model considering the same conditions as in (**D**).

interacts with and is necessary for Oct4 activity (van den Berg *et al*, 2010) and therefore we envision it as an element that confers robustness to the network and thereby to pluripotency, principally through its interactions with Oct4 (see Supplementary Figure S1).

A plethora of recent studies is making increasingly clear that pluripotency does not rely on a single linear program of transcriptional activity, but rather on the convergence of several molecular activities that balance homeostatically self-renewal and differentiation. Here, we have shown that post-transcriptional interactions are an important component of the mechanism that maintains, and perhaps establishes pluripotency. In particular, our results suggest that a noise-driven competitive protein interaction network involving Oct4, Nanog, Tcf3, and β-catenin provide a key element of the regulatory system that maintains pluripotency. In this network, the levels of β-catenin control the stability and dynamics of the different complexes. The impact that protein complexes appear to have on the maintenance of pluripotency places the focus on the need to characterize their biochemical and functional properties. While this work (see also Faunes *et al*, 2013) is a step in that direction, there are technical challenges that will need to be addressed, for example, determining the relative amounts of the different complexes *in vivo* and establishing the impact that these have on the half lives and dynamics of the different components. As our model indicates

that these are important variables in the state of the system, they will have to be addressed experimentally in addition to the more conventional measurements of the transcriptional activity of gene regulatory networks.

## Materials and methods

### Cell culture

Cell lines used are E14Tg2A, EB5, Nanog$^{-/-}$ (Chambers *et al*, 2007) , ZHBTc4 (Niwa *et al*, 2002), AG-23191 (Nishiyama *et al*, 2009), β-catenin$^{+/-}$ and β-catenin$^{-/-}$ (Wray *et al*, 2011), Tcf3$^{+/+}$ and Tcf3$^{-/-}$ (Pereira *et al*, 2006; Yi *et al*, 2008). All mESC lines were cultured on gelatine in Serum and leukemia inhibitory factor (LIF) or on fibronectin in 2i (N2B27, StemCells, Inc., supplemented with 1 μM PD0325901 and 3 μM Chiron) and LIF. MG-132 was used as a proteasome inhibitor at 4 μM concentration. Doxycycline was used to induce the repression of Oct4 expression in the ZHBTc4 cells, or Oct4 overexpression in the AG-23191 cells (for details, see Nishiyama *et al*, 2009 and Faunes *et al*, 2013). Protein half-lives were determined by treating E14Tg2A cells with cycloheximide for 2, 4, 6, or 8 h before total protein extraction for western blot analysis or fixation and immunostaining for quantitative immunofluorescent analysis.

### Immunofluorescence and image analysis

Immunofluorescence, antibodies used, and image analysis were carried out as described previously (Muñoz Descalzo *et al*, 2012).

## RNA-FISH

Fluorescent *in situ* hybridization for Nanog mRNA, imaging and image analysis were performed as previously described (Jakt *et al*, 2013).

## Biochemistry

Nanog mutant cells (cre44) and the parental line (CK044) were cultured in SL or 2i for two passages before lysis in RIPA buffer and concanavilin A fractionation (see Faunes *et al*, 2013 for details), and analysis by western blot using indicated antibodies (see Faunes *et al*, 2013 for details). Quantitative analysis was undertaken using the Licor scanner and software.

To determine the half-lives of proteins, E14Tg2A cells were plated on gelatine in serum + LIF and the next day the medium was changed to serum + LIF + 40 μM cycloheximide to inhibit translation. Cells were lysed using RIPA buffer after 0, 2, 4, and 8 h in cycloheximide. Quantitative western blots were performed by loading equivalent number of cells in each lane for normalization. Specific antibodies to Nanog, Oct4, TCF3, and total β-catenin were used and the half-life was obtained assuming a first-order degradation.

## Supplementary information

## Acknowledgements

We want to thank Jose Silva, Jenny Nichols and Christian Schroeter for comments and discussions, and Leo Sideris for help with some of the colony assays. AMA, PH, and TB are supported by an ERC Advanced Investigator grant to AMA; SMD by a Herchel Smith postdoctoral fellowship; LMJ was supported by Riken intramural funding; FF by postdoctoral fellowship Beca Chile (74100037) and an HFSP award to AMA; PR by the FI program (Generalitat de Catalunya) and JGO by the Spanish Ministerio de Economia y Competividad and FEDER (project FIS2012-37655), and by the ICREA Academia program.

*Author contributions:* SMD carried out the bulk of the experiments and all the quantifications; she also contributed to the discussions on modelling. PR performed the modelling and all the simulations as well as the bioinformatic analysis. SMD and PR worked closely in the critical analysis of the modelling. PH carried out the biochemical analyses and FF measured the half-lives of the different proteins. TB and AMA performed the phenotypic analysis of Nanog mutants and the proteasome inhibitor experiments. LMJ carried out the mRNA counting experiments. JGO contributed to the model and to the calculations. The project was conceived by AMA and JGO in collaboration with SMD and PR. AMA, JGO, SMD, and PR wrote the manuscript.

## Conflict of interest

The authors declare that they have no conflict of interest.

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
