## [Review Process File · Molecular Systems Biology]

A competitive protein interaction network buffers Oct4-mediated differentiation to promote pluripotency in embryonic stem cells

Silvia Muñoz-Descalzo, Pau Rué, Fernando Faunes, Penelope Hayward, Lars Martin Jakt, Tina Balayo, Jordi Garcia Ojalvo, Alfonso Martinez Arias

Corresponding author: Alfonso Martinez Arias, University of Cambridge

Review timeline:

Submission date:	23 April 2013
Editorial Decision:	24 May 2013
Revision received:	19 June 2013
Editorial Decision:	02 August 2013
Revision received:	03 August 2013
Editorial Decision:	20 August 2013
Revision received:	23 August 2013
Accepted:	23 August 2013

Editor: Maria Polychronidou, Thomas Lemberger

Transaction Report:

1st Editorial Decision

24 May 2013

Thank you again for submitting your work to Molecular Systems Biology. We have now heard back from the three referees who agreed to evaluate your manuscript. As you will see from the reports below, the reviewers acknowledge that your work is addressing a potentially interesting topic. However, they raise concerns on your work, which should be convincingly addressed in a revision of the manuscript.

Several of the reviewers' comments refer to the need to provide additional references and better document/clarify several points throughout the manuscript. However, they also point out that additional experimentation is required in order to convincingly support the presented model. Among the more fundamental issues are the following:

- Experimental evidence is required to show the beta-catenin/Nanog interaction in ESCs.
- Due to the key role of protein stability for the model predictions, it is necessary to experimentally validate the predicted complex stabilities.

On a more editorial note, the reviewers refer to the need to carefully consider the use of the term "pluripotency" throughout the manuscript. Additionally, we realize that reviewer #3 suggests that additional discussion could be provided in the supplemental text. However, our policy is to restrict supplementary information to essential data and therefore we would prefer to have the points of this reviewer addressed in the introduction and/or discussion sections of the main text.

If you feel you can satisfactorily deal with these points and those listed by the referees, you may wish to submit a revised version of your manuscript. Please attach a covering letter giving details of the way in which you have handled each of the points raised by the referees. A revised manuscript

will be once again subject to review and you probably understand that we can give you no guarantee at this stage that the eventual outcome will be favorable.

Referee reports:

Reviewer #1:

This study from the Martinez-Arias group is a follow-up study on their previous published work establishing a correlation between the levels of Oct4 and Nanog as a signature for naive pluripotency in mouse ESCs using single cell quantitative immunofluorescence (QIF) technique combined with the Matlab computational tool (Stem Cells 2012). In this study, the authors went a step further by combining single cell QIF and gene expression analysis, together with theoretical modeling, to investigate how the activity of Oct4, Nanog, b-catenin, and Tcf3 is regulated. They found that the activity of these four factors suffices to account for the behavior of ESCs under different conditions. Moreover, their models suggest that the function of the network is to buffer the transcriptional activity of Oct4, that the protein network can explain the mechanism underlying the gain and loss of function in b-Catenin and Tcf mutants.

Several mathematical models of pluripotency have been previously published, which were primarily focused on transcriptional control. However, this cannot reproduce the correlation model the authors previously defined (stem cells, 2012), which prompted the authors to further test the posttranslational regulatory mechanisms (PTMs) and incorporated protein complex regulatory mechanisms into their models. This is largely consistent with the current knowledge and established roles of both PTMs of pluripotency factors as well as protein complexes of Nanog/Oct4 in controlling of stem cell pluripotency. Therefore, the study is logically sound, and it should provide additional thoughts to further test the model and help understand the molecular basis of pluripotency.

While I found the study interesting and may be accepted for publication with revision, I do find a few things need to be further clarified:

1. The main concern I have is the assumption of b-catenin-Nanog complex that authors presented in the study. To my knowledge, there is not a single study has ever proven the existence of beta-catenin/Nanog interaction in ESCs. The authors may have to prove this interaction present in ESCs by coIP/IP studies to make their b:N complex credible in their model.

2. In contrast, Oct4-b/catenin interaction has been demonstrated by published studies. However, several important references documenting such b-catenin/Oct4 association in ESCs were missing and should be added to the references. For example:

Biochem Biophys Res Commun. 2007 Feb 16;353(3):699-705. Epub 2006 Dec 20.

Takao Y, Yokota T, Koide H.

Beta-catenin up-regulates Nanog expression through interaction with Oct 3/4 in embryonic stem cells.

-This study showed that b-catenin physically associates with Oct-3/4.

Oct4 links multiple epigenetic pathways to the pluripotency network.

Ding J, Xu H, Faiola F, Ma'ayan A, Wang J.

Cell Res. 2012 Jan;22(1):155-67. doi: 10.1038/cr.2011.179. Epub 2011 Nov 15.

-This study showed that b-catenin is a partner of Oct4 in an extended Oct4 interactome.

3. The TBON model contains 32 parameters, how the authors know that their result is not simply an overfilling?

4. Do the authors really need in their model that many (unknown) details? Is it possible to simplify the model and reduce the number of parameters by merging/omitting certain equations? Usually that would help.

5. Throughout the paper, whenever a protein complex involving Nanog and Oct4 is mentioned, references such as "van den berg et al, 2010; Wang et al., 2006" were cited, however, the Oct4 interactome paper by "Ding et al. Cell Res 2011" as shown in the point 2 above should also be cited, as that was the only Oct4 interactome among the three published Oct4 interactomes in the current literature that links beta-catenin to the Oct4 network. This is particularly important as O:b is

considered a critical parameter in the models presented in this study.

Reviewer #2 :

This is an interesting and thought provoking manuscript from Descalzo, Rue, and colleagues. The authors note that despite an impressive focus of stem cell researchers on a transcriptional network that controls pluripotency, models lacking protein dynamics provide poor descriptions of the pluripotent state. The manuscript introduces a novel model for the effects of Oct4, Nanog, β -catenin, and Tcf3 in mouse embryonic stem cells. An important element of the model is that it allows for different effects caused by free proteins and those found in complex with other molecules in the system. Another important point is that by including Tcf3 and β -catenin, the model provides a mechanism to account for the significant effects of GSK3-inhibition on mES cells.

One strength of the model is that it explains some recent observations that may otherwise appear to contradict current consensus views. In particular, the model can account for recent work showing pro-differentiation effects of Oct4 and the importance of the ratio of Oct4:Nanog for pluripotency. There are also some weaknesses, and the model is not comprehensive. Many gene products important for self renewal, such as Esrrb, Stat3 and Klf4, are not included. That said, I found the manuscript to be an overall quite compelling description of a new protein based model of pluripotency circuitry. I think it provides a perspective that should be strongly considered as a foundation for subsequent refinements following new discoveries.

I have a number of concerns:

1. Y-axes are not labeled on many graphs (Figure 1G, 2E for example).
2. Knowing the stability of individual proteins in ES cells is an important addition (Fig S3), and is necessary for the key element of the authors' overall model. In addition, knowing how the stability of individual proteins differs when they are bound/not bound by other proteins provides the basis for model. The data shown are for total protein, whether part of a complex or not. Thus, values for complexes are not known, but were inferred from models. Since the stability of complexes is so important to the model, the study should include some validation of predicted stabilities. The half life of proteins in Nanog^{-/-} or β -catenin^{-/-} cells could be validated fairly easily.
3. Figure 4C does not convincingly demonstrate that Nanog^{-/-} ES cells have higher levels of β -catenin and Oct4 proteins than Nanog^{+/+} cells. The problem is that Western blot was unevenly loaded, and therefore not reliable for densitometry.
4. The use of O/N ratio is a bit confusing for analyzing Tcf3 mutants. Overexpression of Nanog does not stimulate differentiation. So, having a poor O/N correlation is detrimental to pluripotency in only one direction. This becomes problematic when analyzing Tcf3^{-/-} in S+L. The cells resist differentiation even though they have a very weak O/N correlation.
5. The manuscript uses the terms pluripotent and pluripotency in an awkward and confusing manner in some places. Figure 4A uses a category of "pluripotent" to describe colony morphology; however, this should just be alkaline phosphatase positive. The manuscript also uses the term "states" of pluripotency, but it is not clear if the authors actually mean naïve and primed states of ESC and EpiSC. Finally, the manuscript sometimes uses the term as if there were different degrees of pluripotency by which one cell is "more" pluripotent than another. By the common definitions of the word - having the ability to contribute to all adult cell types, or to all three germ layers- a cell either is or is not pluripotent. It is rather like how a female either is or is not pregnant. I believe that a more strict use of the term pluripotency would aid in the communication of findings.

Reviewer #3 :

The manuscript by Prof Martinez Arias and co-workers entitled "A competitive protein network buffers Oct4-mediated differentiation promoting pluripotency" combines imaging and biochemical studies of wild type and various mutant mouse ESCs as well as modeling. I support the publication of this manuscript despite the fact that the findings are surprising based upon a number of reports that the authors cite and even those that they do not. I do believe that the computational modeling is well performed and complemented by imaging experiments with mutant cell lines. While there are a number of experimental approaches that could be performed to augment this work, I don't believe that any of them are crucial for publication in MSB and therefore would not hold up the publication of the work for the completion of those experiments. Therefore, I will combine the review and synopsis.

The authors first show that a transcriptional network is unable to account for observed protein expression associated with pluripotency. This should not be a surprise to anyone. There have been many transcriptional networks that have been published, many more than the authors cite or incorporate into their modeling. Therefore, their transcriptional network is not comprehensive. In addition, a number of reports have pointed to post-transcriptional and post-translational components to the pluripotency network. One of the many examples that is not cited in the text is the 2009 Nature Lemischka paper. Thus, while I agree it is useful to verify the holes in transcriptional networks prior to discussing their own models, the authors should not over-stress the deficiencies in these models. I believe by including most of this work in the supplementary text is appropriate.

Next, the authors provide experimental and model simulation for a pluripotency network based on a complex of Oct4 and Nanog. While under some experimental conditions this model accurately predicts the experimental results, the authors point out that this minimalistic model cannot account for all published experimental data. Based on a number of reports, the authors then expand their protein interaction network model to include b-catenin and Tcf3 to the Oct4 and Nanog complex - the TBON network. The authors test the model under different culture conditions and, importantly, using mouse ESCs with genetic manipulations of the various components of the TBON network. The testing of the network is the strongest part of the paper largely because authors, reviewers and editors too often ignore these critical experiments.

The following are not clear in the manuscript:

1. The following is an assumption that is not listed. The transcriptional activity of Oct4 and Nanog is assumed to only be as transcriptional activators. While there is data to support this, many studies have interpreted data that these transcription factors have repressor activity directly or complex with repressors. This assumption needs to be clear to the reader. Furthermore, cite the relevant papers.
2. The authors use the term pluripotential too loosely. Sometimes as far as I can tell, the only evidence is that the colonies are 2i-like.
3. It is not always clear in the text which data was generated by the authors or by the groups which made the mutant ESCs or the simulated data.

This manuscript could be further improved by the following experiments. Without these experiments, the authors MUST be clear that while there is some experimental evidence for their model, additional experiments need to be performed to validate it. These experiments would be required at other journals that cannot appreciate modeling as much as MSB.

1. Perform proximity ligation assay experiments to validate the predicted protein-interaction experiments in single cells. These are straight-forward experiments and complement the quantitative IF experiments that the authors have largely based their work. On bulk populations, co-IPs could be performed.
2. Perform ChIP-qPCR on select targets predicted to have altered TF enrichment based upon the culture conditions used and/or the mutant cell lines used. Complement this with qPCR experiments. This would complement figures 3-6.
3. Assumption iii could be test in terms of the stabilities of the complexes.

4. The quantitative IF experiments would be cleaner and likely more informative if the authors used micropatterning to grow the mESC colonies in a uniform manner. This work was pioneered for the modeling of ESC fate analysis by the Zandstra lab and is being picked up by other groups. However, this technology is not available in many labs and I would never hold up a paper for this type of experiment unless the manuscript could not make conclusions without it.

1st Revision - authors' response

19 June 2013

Reviewer #1 : This study from the Martinez-Arias group is a follow-up study on their previous published work establishing a correlation between the levels of Oct4 and Nanog as a signature for naive pluripotency in mouse ESCs using single cell quantitative immunofluorescence (QIF) technique combined with the Matlab computational tool (Stem Cells 2012). In this study, the authors went a step further by combining single cell QIF and gene expression analysis, together with theoretical modeling, to investigate how the activity of Oct4, Nanog, b-catenin, and Tcf3 is regulated. They found that the activity of these four factors suffices to account for the behavior of ESCs under different conditions. Moreover, their models suggest that the function of the network is to buffer the transcriptional activity of Oct4, that the protein network can explain the mechanism underlying the gain and loss of function in b-Catenin and Tcf mutants. Several mathematical models of pluripotency have been previously published, which were primarily focused on transcriptional control. However, this cannot reproduce the correlation model the authors previously defined (stem cells, 2012), which prompted the authors to further test the posttranslational regulatory mechanisms (PTMs) and incorporated protein complex regulatory mechanisms into their models. This is largely consistent with the current knowledge and established roles of both PTMs of pluripotency factors as well as protein complexes of Nanog/Oct4 in controlling of stem cell pluripotency. Therefore, the study is logically sound, and it should provide additional thoughts to further test the model and help understand the molecular basis of pluripotency.

Reviewer 1 values our combination of experimental data together with theoretical modelling to gain insight into the molecular mechanism involved in the maintenance of pluripotency. The hypothesis of PTM regulating pluripotency is consistent with current knowledge, as transcriptional networks, on their own, have been failing to fully explain how pluripotency is regulated.

While I found the study interesting and may be accepted for publication with revision, I do find a few things need to be further clarified:

This reviewer is very positive to our study, but raises a number of concerns that we address here:

1. The main concern I have is the assumption of b-catenin-Nanog complex that authors presented in the study. To my knowledge, there is not a single study has ever proven the existence of beta-catenin/Nanog interaction in ESCs. The authors may have to prove this interaction present in ESCs by coIP/IP studies to make their b:N complex credible in their model.

We have reported the existence of this complex before (Faunes et al. Development. 2013; 140: 1171-83) and make a more explicit reference to it in the revised version of the ms in the relevant locations (Introduction: p.3; Results: p.4,5,6,7,9,10). The presence of a β :N complex is indeed an important element in the theoretical model and is predicted to be detectable when ES cells are grown the cells in 2i conditions, which is what we observe.

2. In contrast, Oct4-b/catenin interaction has been demonstrated by published studies. However, several important references documenting such b-catenin/Oct4 association in ESCs were missing and should be added to the references. For example: Biochem Biophys Res Commun. 2007 Feb 16;353(3):699-705. Epub 2006 Dec 20. Takao Y, Yokota T, Koide H. Beta-catenin up-regulates Nanog expression through interaction with Oct 3/4 in embryonic stem cells.

-This study showed that b-catenin physically associates with Oct-3/4. Oct4 links multiple epigenetic pathways to the pluripotency network. Ding J, Xu H, Faiola F, Ma'ayan A, Wang J. Cell Res. 2012 Jan;22(1):155-67. doi: 10.1038/cr.2011.179. Epub 2011 Nov 15. -This study showed that b-catenin is a partner of Oct4 in an extended Oct4 interactome.

We appreciate the call on these papers, which we know well and have cited before. We have now incorporated them into the text (Introduction: p. 3, 4; Results: p. 6; and SM).

3. The TBON model contains 32 parameters, how the authors know that their result is not simply an overfilling?

The purpose of our model is to account for as many perturbations as possible of the network formed by Nanog, Oct4, Tcf3 and β -catenin. This requires accounting explicitly for the details involved in these perturbations. The model considers 8 "post-transcriptional" variables (the 4 species and also 4 complexes, O:N, β :N, β :O, β :T) and 4 transcriptional variables (the 4 mRNAs). Out of the 31 parameters mentioned by the reviewer (one of the parameters in Table SII is duplicated to account for the two conditions studied), 15 are directly related to the transcriptional regulation (gene transcription states and mRNA dynamics). These parameter values have been set to match previously published experimental estimates (Fig. S3A) or independently fit to our transcription data (Fig 11). Therefore we do not rely on these parameter values to fit the model to the experimentally measured protein correlations, but only on those related to complex association/dissociation processes and protein and complex degradations. Given the large number of experimental observations used (protein correlations in wild type cells and multiple mutants for S+LIF and 2i+LIF conditions, and total protein degradation rates), we believe we have sufficient experimental constraints to estimate those remaining parameters. Indeed, the parameter values reported here resulted from a systematic investigation of the complete parameter space, which substantially restricted the parameter region compatible with our experimental results. We have now performed a sensitivity analysis of the model that shows a reasonable balance between robustness and specificity, since high correlations persist in the presence of parameter variations of 25% above and below the basal values chosen in the paper. This sensitivity analysis is now included in the Supplementary Material (new Sup. Fig. S7).

4. Do the authors really need in their model that many (unknown) details? Is it possible to simplify the model and reduce the number of parameters by merging/omitting certain equations? Usually that would help.

We did analyse several simplified versions of the network, including the NOC model discussed in detail in the first part of the results ('Oct4/Nanog complex formation as the molecular basis of pluripotency: the NOC model'). This minimal model contains only 3 protein species (Nanog, Oct4 and the Nanog:Oct4 complex) and 5 parameters directly related to these (the complex binding and unbinding rates and the degradation rates). Our results show that this model reproduces the Nanog-Oct4 correlation observed in S+L and 2i+L conditions. However, and crucially, this model cannot reproduce certain well-known established facts, such as the pluripotent character of the Nanog mutant. Similarly, other simplified models were tested and did not stand the comparison with experiments in multiple respects. In particular, we cannot reduce the number of complexes in the TBON model if we want to explain all correlations observed. We now mention explicitly in the text the minimal character of the model proposed.

5. Throughout the paper, whenever a protein complex involving Nanog and Oct4 is mentioned, references such as "van den berg et al, 2010; Wang et al., 2006" were cited, however, the Oct4 interactome paper by "Ding et al. Cell Res 2011" as shown in the point 2 above should also be cited, as that was the only Oct4 interactome among the three published Oct4 interactomes in the current literature that links beta-catenin to the Oct4 network. This is particularly important as O:b is considered a critical parameter in the models presented in this study.

We have incorporated this reference, along with the one by Takao et al. in the new version of the manuscript, when mentioning the existence of a β -catenin/Oct4 complex. See also comment 2 of this reviewer.

Reviewer #2 :

This is an interesting and thought provoking manuscript from Descalzo, Rue, and colleagues. The authors note that despite an impressive focus of stem cell researchers on a transcriptional network that controls pluripotency, models lacking protein dynamics provide poor descriptions of the pluripotent state. The manuscript introduces a novel model for the effects of Oct4, Nanog, β -catenin, and Tcf3 in mouse embryonic stem cells. An important element of the model is that it allows for

different effects caused by free proteins and those found in complex with other molecules in the system. Another important point is that by including Tcf3 and β -catenin, the model provides a mechanism to account for the significant effects of GSK3-inhibition on mES cells. One strength of the model is that it explains some recent observations that may otherwise appear to contradict current consensus views. In particular, the model can account for recent work showing pro-differentiation effects of Oct4 and the importance of the ratio of Oct4:Nanog for pluripotency. There are also some weaknesses, and the model is not comprehensive. Many gene products important for self renewal, such as Esrrb, Stat3 and Klf4, are not included. That said, I found the manuscript to be an overall quite compelling description of a new protein based model of pluripotency circuitry. I think it provides a perspective that should be strongly considered as a foundation for subsequent refinements following new discoveries.

This reviewer values the study presented in this manuscript and how it can explain recent surprising results on the effects on Oct4 levels in pluripotency (Karwacki-Neisius V et al. Cell Stem Cell. 2013; 12(5):531-45 and Radzishheuskaya et al. Nat Cell Biol. 2013;15(6):579-90). We have now included these works as references and discussed them in the text (Results: p.7; Discussion: p. 10), as they have been published since our submission.

While it is true that many more factors are involved in the maintenance of pluripotency (like Esrrb, Stat3 and Klf4 as mentioned by the reviewer), adding them to our theoretical model would add more complexity to it, making it less intuitive and manageable. With our theoretical model we were seeking to have a minimal framework that would explain our experimental observations, including the behaviour of mutants for the key components of the network. Incorporating more factors to our network would involve adding even more parameters to the theoretical model, which already has 32 parameters as pointed out by Reviewer 1. In any case, we did try to account for additional factors, such as Esrrb, at the transcriptional level (see Fig. S1 and the corresponding discussion in the Supplementary Material). These considerations took us to the TBON model as the minimal circuit that explains the cells' behavior. We also make a point about the role of additional elements at the end of the discussion.

Addressing the specific comments of this Reviewer:

1. Y-axes are not labeled on many graphs (Figure 1G, 2E for example).

The label of the y-axis in the graphs pointed out by the reviewer was specified in the Figure Legends (Pearson correlation coefficient). In the new version of the manuscript, we have included a label in the figure as well for an easier interpretation of the results shown in this type of graphs ('Correlation').

2. Knowing the stability of individual proteins in ES cells is an important addition (Fig S3), and is necessary for the key element of the authors' overall model. In addition, knowing how the stability of individual proteins differs when they are bound/not bound by other proteins provides the basis for model. The data shown are for total protein, whether part of a complex or not. Thus, values for complexes are not known, but were inferred from models. Since the stability of complexes is so important to the model, the study should include some validation of predicted stabilities. The half life of proteins in Nanog^{-/-} or β -catenin^{-/-} cells could be validated fairly easily.

We have now included in the new version of the manuscript the half-lives of Oct4 and Nanog under specific culture conditions and mutant backgrounds (Fig. 2H). We observe an increase in the half-lives of both proteins in 2i+LIF conditions, agreeing with the idea that under these conditions, there is a higher amount of protein found in the complexes (see also Faunes et al., Development 2013;140(6):1171-83) that would stabilize them, extending their half-lives. Nanog half-life decreases in the absence of Oct4 (ZHBTC4 cells treated with Dox), consistent with the idea that Oct4 contributes to the stabilization of Nanog. In the absence of β -catenin (and in 2i+LIF conditions) there is no change in the Nanog half-life.

In our results we observe that while β -catenin contributes to the stability of Oct4, Nanog does not, as assumed by our theoretical model.

3. Figure 4C does not convincingly demonstrate that Nanog^{-/-} ES cells have higher levels of β -catenin and Oct4 proteins than Nanog^{+/+} cells. The problem is that Western blot was unevenly loaded, and therefore not reliable for densitometry.

Whilst we do agree that the samples are not evenly loaded in this WB (as reflected by the tubulin), the differences are all within 2 order of magnitude and none of the pixels were saturated. The LICOR system used in this analysis is quantitative over 3.6 orders of magnitude. Therefore although not best practise, the loading of this blot does not compromise our ability to quantify the relative amount of each protein in the samples. Whole blots, and actual measurement figures can be supplied.

4. The use of O/N ratio is a bit confusing for analyzing Tcf3 mutants. Overexpression of Nanog does not stimulate differentiation. So, having a poor O/N correlation is detrimental to pluripotency in only one direction. This becomes problematic when analyzing Tcf3^{-/-} in S+L. The cells resist differentiation even though they have a very weak O/N correlation.

The reviewer is right in that the O/N the correlation descriptor is not enough to characterise pluripotency, but it is a good indicator. We have discussed the state of pluripotency in the absence of Tcf3 in a previous study (Muñoz-Descalzo et al., Stem Cells. 2012;30(12):2683-91). As we discussed in that study, not only the O/N correlation coefficient is a quantifier of the pluripotency, but also the O/N ratio of the cell population (given, for instance, by the slope of a linear regression). In Tcf3 mutant cells, the O/N correlation decreases, suggesting that they are not in a stable pluripotent state; however the O/N ratio is also reduced, indicating that these cells have, on average, lower levels of Oct4 and/or higher levels of Nanog as compared to the wild type (which they actually do, see new Fig. S14B-C) and thus they might resist Oct4-mediated differentiation. According to our model, this can be explained by the absence of "free" Oct4 in those situations (see Fig. 5D) that would promote differentiation.

5. The manuscript uses the terms pluripotent and pluripotency in an awkward and confusing manner in some places.

We have tried to be more precise in the way we use with these terms in the new version of the manuscript. In the Introduction we are now more explicit about what does 'pluripotency' mean.

Figure 4A uses a category of "pluripotent" to described colony morphology; however, this should just be alkaline phosphatase positive.

The description of colony morphology in our colony-forming assays sought to be clear and obvious for a non-specialist in the ESC field interested in reading this manuscript. Although 'alkaline phosphatase positive' (or 'negative') as suggested by the reviewer would be more precise terminology, we have decided to use 'differentiated' for alkaline phosphatase negative colonies and 'undifferentiated' for the positive colonies, as we believe these are both precise and understandable for non-specialists.

The manuscript also uses the term "states" of pluripotency, but it is not clear if the authors actually mean naïve and primed states of ESC and EpiSC.

Finally, the manuscript sometimes uses the term as if there were different degrees of pluripotency by which one cell is "more" pluripotent than another. By the common definitions of the word - having the ability to contribute to all adult cell types, or to all three germ layers- a cell either is or is not pluripotent. It is rather like how a female either is or is not pregnant. I believe that a more strict use of the term pluripotency would aid in the communication of findings.

As mentioned above, we tried to be clearer in the use of the term 'state of pluripotency' in the new version. Since our experimental system is mouse ESC we are not extrapolating our protein network to EpiSC, as both culture requirements and probably the relation among network components might be different.

Reviewer #3 (Remarks to the Author): The manuscript by Prof Martinez Arias and co-workers entitled "A competitive protein network buffers Oct4-mediated differentiation promoting pluripotency" combines imaging and biochemical studies of wild type and various mutant mouse ESCs as well as modeling. I support the publication of this manuscript despite the fact that the findings are surprising based upon a number of reports that the authors cite and even those that they do not. I do believe that the computational modeling is well performed and complemented by imaging experiments with mutant cell lines. While there are a number of experimental approaches that could be performed to augment this work, I don't believe that any of them are crucial for

publication in MSB and therefore would not hold up the publication of the work for the completion of those experiments. Therefore, I will combine the review and synopsis.

We appreciate the observation of the reviewer that our manuscript should not be seen, simply, as an experimental piece of work and that this makes less imperious the need for some (difficult) experiments.

The authors first show that a transcriptional network is unable to account for observed protein expression associated with pluripotency. This should not be a surprise to anyone. There have been many transcriptional networks that have been published, many more than the authors cite or incorporate into their modeling. Therefore, their transcriptional network is not comprehensive. In addition, a number of reports have pointed to post-transcriptional and post-translational components to the pluripotency network. One of the many examples that is not cited in the text is the 2009 Nature Lemischka paper. Thus, while I agree it is useful to verify the holes in transcriptional networks prior to discussing their own models, the authors should not over-stress the deficiencies in these models. I believe by including most of this work in the supplementary text is appropriate.

While the transcriptional model presented in the Supplementary Material is not comprehensive, as stated by the reviewer, we do think that it gives an insight into the limitations of transcription regulation models. As indicated above (reply to reviewer 2) we have tried to reason our reduction of a larger and more comprehensive network on the basis of what is essential to reproduce our experimental observations. We should also point out that it remains to be seen if adding more elements to the network improves the fit. One of the strengths of our model, we believe, is that, in contrast with other models which try to capture a loosely defined pluripotency term, ours targets the experimental distributions. This constrains what we can do, but on the other hand justifies our using the simpler possible model to explain the facts (see responses to reviewers 1 and 2 for modelling complexity).

We now take into account and discuss the Lemischka reference (Lu et al. Nature. 2009;462(7271):358-62). We have also included the papers mentioned by Reviewer 1 (Takao et al. Biochem Biophys Res Commun. 2007; 353(3):699-705 and Ding et al. Cell Res. 2012;22(1):155-67) which demonstrate the existence of a β :O complex and also Sanchez-Ripoll et al. (PLoS One. 2013;8(4):e60148), which supports a translational regulation of pluripotency.

Next, the authors provide experimental and model simulation for a pluripotency network based on a complex of Oct4 and Nanog. While under some experimental conditions this model accurately predicts the experimental results, the authors point out that this minimalistic model cannot account for all published experimental data. Based on a number of reports, the authors then expand their protein interaction network model to include b-catenin and Tcf3 to the Oct4 and Nanog complex - the TBON network. The authors test the model under different culture conditions and, importantly, using mouse ESCs with genetic manipulations of the various components of the TBON network. The testing of the network is the strongest part of the paper largely because authors, reviewers and editors too often ignore these critical experiments.

We appreciate that the Reviewer acknowledges our effort trying to test whether the model can be useful by testing it in the different conditions: a model that is not tested for its robustness with experiments cannot be useful to understand the reality that it is trying to model.

The following are not clear in the manuscript: 1. The following is an assumption that is not listed. The transcriptional activity of Oct4 and Nanog is assumed to only be as transcriptional activators. While there is data to support this, many studies have interpreted data that these transcription factors have repressor activity directly or complex with repressors. This assumption needs to be clear to the reader. Furthermore, cite the relevant papers.

We thank the reviewer for pointing out this source of confusion. We do not assume any particular type (positive or negative) of transcriptional activity for Nanog and Oct4 in the model or data analysis. In particular, the bioinformatic analysis utilises data from ChIP-seq and Gene expression (Loss/Gain of Function) from several studies (see Table SIII in the Supplementary Material). The ChIP data provides evidence only on the direct protein-DNA interactions, whereas Loss and Gain of function data gives information on functionality. In the latter case (LoF and GoF), however, we consider those genes that are differentially expressed as possible Nanog or Oct4 targets irrespective

of whether their expression is increased or reduced with respect to the control. We now have made this clear in the supplementary material and have rephrased some sentences in the main text and SM to avoid confusion.

2. The authors use the term pluripotential too loosely. Sometimes as far as I can tell, the only evidence is that the colonies are 2i-like.

Reviewer 2 made a comment to the same effect. We have now tried to be more careful in our description of pluripotency in the new version of the manuscript. Regarding the '2i-like' colonies, we use this term to morphologically describe compact, tight, ball-shaped colonies similar to the one that ESCs acquire when grown in 2i+LIF conditions; therefore we associate that this colony morphology is adopted by cells that are mainly in the ground state of pluripotency. We have included this description in the text (SM p. 17). In our view, colony-forming assays have a wealth of information which is often disregarded in publications and we are trying to make this explicit.

3. It is not always clear in the text which data was generated by the authors or by the groups which made the mutant ESCs or the simulated data.

We have tried to clarify it in the new version of the manuscript changing the location of where the references are inserted in the text when we refer to data generated by other authors. For the simulated data, and to distinguish them from the experimental data, we did include a 'vanilla' background in the graphs.

This manuscript could be further improved by the following experiments. Without these experiments, the authors MUST be clear that while there is some experimental evidence for their model, additional experiments need to be performed to validate it. These experiments would be required at other journals that cannot appreciate modeling as much as MSB.

We appreciate the understanding of this Reviewer in that the additional experiments suggested, although very valuable and informative, are out of the scope of the current study. In terms of the suggested consideration we have made it explicit in the final paragraph of the discussion.

1. Perform proximity ligation assay experiments to validate the predicted protein-interaction experiments in single cells. These are straight-forward experiments and complement the quantitative IF experiments that the authors have largely based their work. On bulk populations, co-IPs could be performed.

While it is true that proximity ligation assay experiments are a very powerful tool that would allow the quantification at the single cell level of protein interactions as described in Gajadhar & Guha (Biotechniques.2010;48(2):145-52) for the dimerization of the EGF receptor, we do not feel that this technique is so straightforward as the Reviewer suggests as it involves the design, preparation and optimization of probes. On bulk populations, we and others have already shown the presence of the β :O, β :N, β :T and O:N complexes and hope that this will suffice for now. We also hope that our work will stimulate this type of assays and experiments in a field which is too based on transcriptional information. We certainly are pursuing this.

2. Perform ChIP-qPCR on select targets predicted to have altered TF enrichment based upon the culture conditions used and/or the mutant cell lines used. Complement this with qPCR experiments. This would complement figures 3-6.

There are two already published studies (one of them since the submission of our study) that show ChIP data under genetic backgrounds seeking for transcriptional targets of Oct4 and/or Nanog. Specifically, the Oct4-Nanog binding sites in wild-type cells were described in Chen et al. (Cell. 2008; 133:1106-17), and these are the data used in our bioinformatics analysis as indicated in the Supplementary Information (Table SIII).

There are also recent ChIP data on the different mutant cell lines. Nanog mutant cells ChIP data were published in Navarro et al. (The EMBO Journal. 2012; 1-16). In this study, the authors use the results to show the existence of an auto-regulatory negative feedback loop of Nanog onto its own promoter. The same group recently published ChIP data in cells with different levels of Oct4 (Karwacki-Neisius V et al. Cell Stem Cell. 2013; 12(5):531-45). Since the data suggested by the

reviewer have been already published we feel that it is not necessary to repeat them.

3. Assumption iii could be test in terms of the stabilities of the complexes.

We agree that this would be an ideal and significant experimental result, however, this is not a trivial experiment. We are pursuing this but, at present, do not have much to show for it. The one experiment one might think of doing (and we have considered it) would involve performing pull downs for the different proteins in the presence or absence of cyclohexamide (to inhibit protein synthesis) and then comparing changes over time with regard to the total pools in lysates. We would expect that if indeed the proteins are more stable in the complex vs the free pools, blocking protein degradation will reveal it. This means that were we to see any of the proteins in the pull-down decreasing upon cyclohexamide treatment, slower than in the lysate, this would tell us is that the protein is more protected from degradation when found in the complex and, therefore, more stable. But, a slower decrease should also be interpreted with caution as there could be secondary effects and we might be measuring a side effect. Furthermore, the pull downs are not quantitative, we do not yet know how much of the total fraction of any of the proteins is in pull downs (we can get a number but are not sure about its meaning) and, we always get less of the protein in the complexes (something normal in these experiments). Thus any change in the half life of any protein could –and probably should- be interpreted as reflecting the relative amounts of starting materials rather than their actual half life. We are thinking about these problems trying to find a way around them but it is not easy as it requires sophisticated biochemistry which we believe is beyond the scope of this work, as reviewer 3 acknowledges explicitly.

One thing that we have done is to measure the half life of the different proteins in the absence of each other in whole lysates and have included and discuss this information in the manuscript (new Fig. 2H). See also the response to point 2 from reviewer #2.

4. The quantitative IF experiments would be cleaner and likely more informative if the authors used micropatterning to grow the mESC colonies in a uniform manner. This work was pioneered for the modeling of ESC fate analysis by the Zandstra lab and is being picked up by other groups. However, this technology is not available in many labs and I would never hold up a paper for this type of experiment unless the manuscript could not make conclusions without it.

We agree with the reviewer that performing QIF on uniformly grown mESC colonies will be very informative and we shall use it in the future.

Additional Correspondence (editor)

18 July 2013

I am contacting you regarding your manuscript MSB-13-4492R. We have now heard back from the reviewers and they are rather supportive. However, Reviewer #1 is not convinced by the biochemical evidence of beta-catenin and Nanog interaction. The reviewer mentions that "the previous study study only provides the fractionation of membrane bound-proteins from soluble ones showing Nanog is present in both membrane bound and soluble complexes containing beta catenin and Oct4 by western blots. It is still not clear whether beta-catenin can form two distinct protein complexes: the beta/catenin-Oct4 and beta/Catenin-Nanog complexes in ESCs, as Nanog association with beta catenin may be indirectly mediated by Oct4". The reviewer recommends to perform coIP/IP experiments for the beta-catenin and Nanog interaction in this study to bolster the existence of the b:N complex in ES cells.

Before making a formal decision, I wanted to discuss with you on whether this important point could be addressed within a reasonable timeframe.

Regarding your suggestion to provide additional simulations in the context of the two recent studies in Cell Stem Cell, we feel that inclusion of additional data is not required at this point.

Thank you very much for your interest in the work and for contacting us on the matter of the Nanog: β -catenin complex and its incidence on the manuscript as things stand at the moment. We found the conversation helpful and believe that the outcome will be an improved manuscript.

Looking at our figure 2, we appreciate the implications of what is depicted in panel B. However as we pointed out and we elaborate here, this complex is not a major driver of the model. This is obvious if one looks at the situation quantitatively and not (as we biologists often do) qualitatively. There is indeed a mention of a B:N complex but this is only present in 2i and accounts for about 5-10% of Nanog. Furthermore, removal of this complex from the model has little effect on the outcome and dynamics, and a model with a ternary complex O:B:N works as well as the current one. The reason why the B:N was included in the final model was to account precisely for observations of the B:N correlations and for the ease of calculations. Having said this, the model does not rule out that this interaction is mediated through the interaction of Oct4 with β -catenin. As we have emphasized, the real driver is the O:B complex, as demonstrated in the experiments and simulations of the Nanog mutants. We shall be happy to qualify all this in the main text with an appropriate reference to both the effect of removal of the complex and the ternary complex model in the supplementary material.

Having said this about the model, we shall also be happy to consider doing the experiment of the Nanog IP. However, after our discussion and in the context of the quantitative considerations we have made above, we would invite you to think whether this extra work (which we are happy to try but will involve time and money) is either revealing or necessary. First of all, the predicted percentage of Nanog in this complex in 2i (the model only predicts its presence in 2i and we only find the interaction in 2i) is very small, which means that it is likely to be difficult to detect (as already suggested by the data that we have of the amount of Nanog present in the complex when pulled down with β -catenin). Second its contribution to the actual central parameter of the system i.e. the amount of free Oct4, is not as large as the O:B complex, which is, as we pointed out, the real driver of the process. Third it might well be that this interaction is, as we have discussed before, brought about by the interaction between Oct4 and Nanog. So, in our view the chances of seeing an exclusive O:B are low and if one sees it one cannot rule out a role for Oct4 in bridging it. The only possible way to test this interaction rigorously would be to do an IP in an Oct4 mutant cell but, as we pointed out earlier, this is not possible because the cells will differentiate and downregulate Nanog and lose their pluripotency.

While as you can see we appreciate the point that has been brought up which has made us all think a bit deeper about the model (and in this sense the reviewing process has very much fulfilled its role), it would be easy to lose sight of the relative importance of each complex and the small contribution that the B:N complex makes to the stability of the system (in vivo and in silica). The main point that the work makes is that the amount of Oct4 is the key to pluripotency and that this is dynamically buffered in complexes with Nanog and, most significantly, with β -catenin. We and the reviewers have underlined this and we shall not elaborate here. This, we think, is clear in the text but will be happy to put this into sharper contrast.

In summary, what we propose is

1. To explain better the way we think and deal with the complexes and how they relate to the biochemistry. Particular mention of the small contribution of the B:N to the system and show it in numerical simulations.
2. To point out that a model with a ternary complex supports the observed distributions as well. We shall add this model in the supplementary.
3. Happy to try the Nanog IP but we would appreciate your feedback on this since, all things considered we are not sure what valuable or additional information this will add to the work (recall the low percentage in the model, the little contribution it makes to the dynamics of the system, and the fact that it only appears in 2i). Furthermore, the experiment will always have to be done in the

presence of Oct4 which, given that Oct4 interacts with both β -catenin and Nanog, will not eliminate the caveat of an indirect interaction as Oct4 will always come down with Nanog. We shall be happy to discuss these matters in Materials and Methods or SM.

We appreciate your feedback but to gain some time, will begin to revise the manuscript along these lines. The IP, as we have discussed and explained here, would and does not have a significant impact on the work.

Additional correspondence (editors)

28 July 2013

Thank you very much for the further clarification over the phone. We think we have now better understood your point of view and the emphasis of the paper on the importance of Oct4 buffering via β -catenin.

Having said that, we think that it is important that the presentation of the study remains clear and coherent.

If the TBON model remains presented as considering "8 protein species: 4 free proteins (O, N, β and T) and 4 protein complexes (O:N, β :N, β :O, β :T)", which "compete for their individual components" (current Fig 2B and proposed thumbnail), then experimental evidence for the existence of these complexes would be expected. We have consulted with Reviewer #3, who felt this would indeed be important data to include in the study.

This may however be difficult, for the reasons you explained in our phone call and below.

We understand that the hypothetical β :N complex may not play a major role and that a model that does not assume the existence of this complex but uses a β :O:N ternary complex may also work (let's transiently call such a model 'TBON3'...). In absence of direct evidence for β :N, such a TBON3 model would appear as more 'parsimonious' than TBON, at least from an experimental point of view.

In fact, the question then arises of whether a model that assumes only the {O:N, β :O, β :T} binary complexes and no β :N complex would work as well. Are β :N or a ternary complex needed at all? If yes, this should be explained.

If a more parsimonious model works fine (which remains to be shown in the revision, of course), it would then be important to present it in the main text as the main model and adapt the presentation of the results in the text and the figures (eg. the β :N complex is referred to at many places) and Supp Info accordingly.

Our worry is that it may become very confusing for readers (as we all were!) to present the TBON model as it is currently depicted in Fig 2B but then include almost as a 'side note' that in fact an alternative model (eg. TBON3 or even a simpler model) that requires less biological assumptions works equally well. In our opinion, the most parsimonious model should be the one that is presented and analyzed in the main text. It is not completely clear to us that whether this can be achieved in only a couple of days, since the computational analysis and the figures would need to be redone with the simplified model, but please let us know if we have misunderstood something.

2nd Editorial Decision

02 August 2013

Thank you again for submitting your work to Molecular Systems Biology. We have now heard back from the three referees who accepted to evaluate the study. As you will see, the referees feel that most of their main concerns have been addressed. However, Referee #1 expresses concerns regarding the existence of the β -catenin/Nanog complex, which we would ask you to carefully address in a revision of the present work.

Referee reports:

Reviewer #1:

The authors addressed most of my initial concerns except one: the biochemical evidence of beta-catenin and Nanog interaction. Although a previous paper was referred to for the potential association of beta-catenin and Nanog, that study only provides the fractionation of membrane bound-proteins from soluble ones showing Nanog is present in both membrane bound and soluble complexes containing beta catenin and Oct4 by western blots. It is still not clear whether beta-catenin can form two distinct protein complexes: the beta/catenin-Oct4 and beta/Catenin-Nanog complexes in ESCs, as Nanog association with beta catenin may be indirectly mediated by Oct4. Although authors mentioned that the b:N complex is predicted to be detectable when ES cells grown in 2i condition, the coIP/IP experiments can be easily done for beta-catenin and Nanog in either culture condition. Such coIP/IP experiments have been performed by several groups for the beta-catenin and Oct4 interaction/association, and similar studies should be performed also for the beta-catenin and Nanog interaction in this study to bolster the existence of the b:N complex in ES cells.

Reviewer #2 :

I believe the authors have satisfactorily addressed the criticisms that I made in the previous round of review. I have no new concerns.

Reviewer #3 :

The authors have done a good job clarifying confusing aspects of the paper. Their work is very timely as there have been a number of papers published this year that could have been predicted by their model or at least a model with a similar framework. This paper should be accepted as is. I will point out though that PLA experiments are very straight-forward. No one in my lab has ever failed in a PLA experiment, using many different probes. This would have added you their paper.

2nd Revision - authors' response

03 August 2013

The early hints over the weekend have been confirmed by the more detailed calculations, and the change of the model from TBON to TBON3 has not altered noticeably the behaviour of the networks. This has made the revision simple. The main reasons for this are, as we discussed, the small contribution of a fl- catenin:Nanog complex to the system and the way in which the different binary complexes lead to a time average equilibrium that matches that of the ternary complex in the model.

We also are pleased to see that a main feature of the model, the prediction of mutants from the wild type parameters, still holds.

Find attached a version of the ms with the most significant changes introduced by the change of model highlighted/tracked. The new model is still called TBON. From the perspective of the flow of the manuscript, the most significant change in the main text is the introduction of the model with the ternary complex up front. We leave the details to the supplementary (also attached with the relevant changes also highlighted/tracked) where we now compare the model with its alternatives: a model without the fl:N complex and the earlier model in which the ternary complex was emulated by the binary complexes. We have kept the discussion of these additional models to a minimum and have avoided showing the data not to add more than is necessary. Happy to include them if you wished but we rather keep things simple and answer direct queries from readers. As you will see because of the formal equivalence of the two models the changes are minimal.

Naturally, figures and their legends have been changed to account for the revised model and the new simulations which, though in essence remain the same, have their own quantitative features. In the main text the changes affect Figs 2 (where the new model is made explicit) and 4-6 and in the supplementary Figs S6-8 and S11-S18.

On the issue of the Biochemistry, we feel that the published evidence for the different complexes discussed should be enough as in most instances there is more than one independent observation: Iping with Oct4 we obtain fl-catenin and Nanog and Iping with fl-catenin we recover Oct4 and Nanog. Nonetheless, we have spent the last few days on the Nanog IP and have the result that we attach which shows that IPing with Nanog, we do obtain fl-catenin and Oct4 (see below). This does not remove the caveats that we have discussed and shall be happy to present and discuss it in the SM, though we feel that a detailed discussion of the cell and molecular biology of the system which might be required to understand these experiments and which refers to the Faunes et al 2013 reference- is out of the scope of this work.

Hopefully you will now find that the manuscript is in a satisfactory condition for publication. My colleagues and I have certainly appreciated the review process which has been in this case very constructive. We also feel that the work is important because it not only provides a useful novel perspective to think about pluripotency but we also believe that the model, by dealing with the behaviour of individual cells and their extrapolation to populations, has several novel aspects that go beyond the continuous dynamical systems that are often used to represent circuits.

Nanog immuno-precipitation was carried out following the protocol described in Faunes et al (2013) Development, using 10 μ l anti-Nanog for the immuno-precipitation reaction. NB The Oct4 band is partly masked by the HlgG used in the IP.

3rd Editorial Decision

20 August 2013

Thank you again for submitting your work to Molecular Systems Biology. We have now heard back from the referee who accepted to evaluate your revised manuscript. As you will see, the referee is now satisfied with the modifications made. However, this referee lists a series of minor points that we would like to ask you to carefully address in a revision of the manuscript.

Please resubmit your revised manuscript online, with a covering letter listing amendments and responses to each point raised by the referees. Please resubmit the paper ****within one month**** and ideally as soon as possible. If we do not receive the revised manuscript within this time period, the file might be closed and any subsequent resubmission would be treated as a new manuscript. Please use the Manuscript Number (above) in all correspondence.

 Referee reports:

Reviewer #1:

somewhat downplayed the contribution of b-catenin:Nanog complex to the system. Although the coIP data quality presented in the rebuttal letter is not ideal, the efforts the authors devoted to resolve the issue is commendable. I believe the revised manuscript has now met the publication standard. The following minor points are largely stylistic for clarity in the final print.

Minor points:

1. In page 3, line 3, "," should be inserted before and after "i.e."
2. There are places that references are mis-cited. For example: page 4, last line; page 6, 2nd paragraph, line 8, and last line of page 6. In all these places, the "Fidalgo et al., 2012" reference should be replaced with "Ding et al., 2012". Only the latter paper described and discussed the potential partnership between Oct4 and Nanog.
3. Page 6, the 2nd paragraph from the bottom: the reference between "Wray et al., 2010" and "Lanner et al., 2010" should be separated by ";", but not ")(".
4. Page 9, 2nd paragraph, line 5: while the text says "...of the parental heterozygous cell line (Fig. 5A and S12K)", the Fig. 5A shows the data of WT but not heterozygous. So this needs to be clarified.
5. Page 10, line 5, there should be only one "." but not "..".
6. Page 11, last line of the main text: why "Gene Regulatory Networks" are capitalized for each word?
7. Page 12, "Biochemistry:"- no ":" is necessary.
8. Both in the images of Fig. 4D and Fig. 5A and C: there are many black dots in the images. What are they?

3rd Revision - authors' response

23 August 2013

As suggested by the reviewer, these are the changes in the current version of the manuscript

1. In page 3, line 3, "," should be inserted before and after "i.e.". It seems to us that the reviewer wants us to write ,i.e., and we do not think this is correct so we have left it i.e.
2. There are places that references are mis-cited. For example: page 4, last line; page 6, 2nd paragraph, line 8, and last line of page 6. In all these places, the "Fidalgo et al., 2012" reference should be replaced with "Ding et al., 2012". Only the latter paper described and discussed the potential partnership between Oct4 and Nanog. This has now been corrected in the manuscript.
3. Page 6, the 2nd paragraph from the bottom: the reference between "Wray et al., 2010" and "Lanner et al., 2010" should be separated by ";", but not ")(".
4. Page 9, 2nd paragraph, line 5: while the text says "...of the parental heterozygous cell line (Fig. 5A and S12K)", the Fig. 5A shows the data of WT but not heterozygous. So this needs to be clarified. The figure comparing wt, heterozygous and mutant -catenin results is shown in Fig. S12K, while Fig. 5A shows the comparison between wt and mutant (for consistency with the results shown in Fig. 5C). This has now been corrected in the manuscript.

5. Page 10, line 5, there should be only one "." but not "..".
This has now been corrected in the manuscript.

6. Page 11, last line of the main text: why "Gene Regulatory Networks" are capitalized for each word?
We have now change this to lower case letters.

7. Page 12, "Biochemistry:"- no ":" is necessary.
We have corrected this

8. Both in the images of Fig. 4D and Fig. 5A and C: there are many black dots in the images. What are they?
Not sure what the reviewer refers to, as we do not see them in our files. Maybe he/she is referring to the axis tick marks or to something in her/his viewer or version